# Experimental Investigation on Ablation Behaviors of CFRP Laminates in an Atmospheric Environment Irradiated by Continuous Wave Laser

**DOI:** 10.3390/polym14235082

**Published:** 2022-11-23

**Authors:** Yongqiang Zhang, Jinwu Pan, Shuhan Zhou, Qianfeng Yin, Jialei Zhang, Wenbo Xie, Fuli Tan, Wei Zhang

**Affiliations:** 1Institute of Fluid Physics, China Academy of Engineering Physics, Mianyang 621900, China; 2Department of Astronautic Science and Mechanics, Harbin Institute of Technology, Harbin 150001, China; 3Airport Planning and Design Research Institute, PowerChina Kunming Engineering Corporation Limited, Kunming 650051, China

**Keywords:** laser irradiation, CFRP laminates, ablation behaviors, atmospheric environment

## Abstract

In order to understand the ablation behaviors of CFRP laminates in an atmospheric environment irradiated by continuous wave laser, CFRP laminates were subjected to a 1080-nm continuous wave laser (6-mm laser spot diameter), with different laser power densities carried out in this paper. The internal delamination damage in CFRP laminates was investigated by C-Scan. The rear- and front-face temperature of CFRP laminates were monitored using the FLIR A 655 sc infrared camera, and the rear-face temperature was monitored by K type thermocouples. The morphology of ablation damage, the area size of the damaged heat affected zone (HAZ), crater depth, thermal ablation rate, mass ablation rate, line ablation rate, etc., of CFRP laminates were determined and correlated to the irradiation parameters. It is found that the area size of the damage HAZ, mass ablation rate, line ablation rate, etc., increased as the laser power densities. The dimensionless area size of the damaged HAZ decreased gradually along the thickness direction of the laser irradiation surface.

## 1. Introduction

Carbon fiber reinforced plastics (CFRP) composites are widely used as engineering components in the aerospace, marine, civil or other engineering industries due to their excellent mechanical characteristics at room temperature, compared to common materials [1,2,3,4]. In particular, CFRP has great advantages in military applications, such as unmanned air vehicles (UAVs), which play an important role on the battlefield. Laser weapons have the advantages of high precision, high energy and low cost compared to traditional kinetic energy interception, which can destroy the UAVs. Therefore, the interaction between laser and material has been noted by many researchers. The ablation mechanism of CFRP composites subjected to laser irradiation is relatively complex; it is difficult to effectively characterize the ablation damage mechanism and behavior of CFRP composites. The experiment is still an effective method by which to study the ablation failure mechanism and behavior of CFRP composites under thermal load. However, intense heating produced by the laser can adversely affect the integrity of composite structures [5]. Compared with metal materials, composite materials are made of at least two different components, which will have a more complex response to the laser. When the temperature reaches a specific value, the pyrolysis of epoxy happens. For example, thermal analysis and ablation law of epoxy were discussed in [6,7,8]. At higher temperatures, the composite materials will turn into the process of sublimation. The complex response of composite to laser has a relation with some factors, including laser parameters [9,10,11,12,13,14], such as power density, spot size and irradiation time; characters themselves [13], such as fiber layup sequence and thickness; and environment conditions [15,16,17,18,19,20,21,22,23,24,25,26,27,28,29,30,31,32], such as vacuum, atmosphere environment and tangential flow.

The thermodynamics of CFRP and thermo-mechanical damage behavior need to be further investigated to understand these materials’ vulnerability. The delamination damage and failure conditions of composite materials induced by laser ablation have not been studied deeply.

In this paper, the tests of CFRP laminates subjected to a 1080-nm continuous wave laser (6-mm laser spot diameter) with different laser power densities were carried out to understand the ablation behaviors of composites in atmospheric environment. The reflectance and transmittance of the sample were measured by UV-VIS Absorption Spectrometer. The ablation morphologies of CFRP laminates were measured by VHX-100 digital microscope. The internal delamination damage in CFRP laminates was investigated by C-Scan. The rear- and front-face temperature was monitored using the FLIR A 655sc infrared camera, and the rear-face temperature of sample was monitored by K-type thermocouples that were also monitored. The morphology of ablation damage of CFRP laminates, the area size of the damage heat affected zone (HAZ), crater depth, thermal ablation rate, mass ablation rate, line ablation rate, etc., were presented and analyzed.

## 2. Materials and Specimens

The material used in this study is a unidirectional CFRP composite laminate made by Harbin FRP Institute, China, from commercial prepreg material produced by Toray Industrial Corporation. The carbon fiber is T700, and the epoxy matrix is TDE-85. It is 2-mm thick and consists of 16 layers with quasi-isotropic layering [0/45/90/-45]_2s_. This configuration is widely used when there is no preferential loading direction, which makes laminate useful in many applications. The surface of the prepared CFRP plate is flat and meets the requirements of the laser ablation test. The material properties of CFRP are provided by the manufacturer as shown in Table 1.

Figure 1 is the cross section micrograph of the original target sample prepared at two magnifications in the direction of thickness. The cutting surface is parallel to the fiber direction at the outermost layer. Carbon fiber diameter is about 5 μm, fiber arranged along a direction. Note that the loading axis is parallel to the page in this SEM micrograph. Square and circular CFRP specimens were prepared, as shown in Figure 2. The size of square CFRP specimens was 50 mm × 50 mm, and the diameter of circular CFRP specimens was 70 mm.

## 3. Experimental Equipment and Measuring Systemtimes

All laser experiments were conducted in the high-velocity impact dynamics laboratory at the Harbin Institute of Technology. The experimental setup is shown in Figure 3. CFRP specimens were irradiated by CW laser of different power densities in the atmospheric environment. This is because severe combustion may occur when the CFRP composite is subjected to high heat flux in this environment. A Ruike 1080-nm fiber laser of 1 kW provided by IPG Ltd. was utilized as the laser source. The laser beam was irradiated on the center of the testing zone with a fixed diameter of 6 mm, and the power density can be controlled by changing the laser power output. In the experiments, samples were irradiated by laser powers of 250 W, 500 W, 750 W and 1000 W for the 20 s, respectively. The reflectance and transmittance of the sample were measured by UV-VIS Absorption Spectrometer. The ablation morphologies of CFRP laminates were measured by VHX-100 digital microscope. The internal delamination damage in CFRP laminates was investigated by C-Scan technique. The rear- and front-face temperatures were monitored using the FLIR A 655sc infrared camera; K type thermocouples also monitored the rear-face temperature.

## 4. Results and Discussion

### 4.1. Optical Performance

Laser is converted into thermal energy by thermal coupling with the material. The reflectance and transmittance of the material to the laser directly affect the magnitude of the thermal coupling coefficient. Therefore, the reflectance and transmittance of the sample were measured by UV-VIS Absorption Spectrometer.

In order to accurately measure the reflection and transmission properties of the CFRP laminates, the front and rear surfaces of the CFRP laminates were wiped with alcohol cotton before measurement.

Figure 4 shows the variation curves of transmittance and reflectance of CFRP laminates as a function of incident wavelength. It can be seen, from Figure 4a, that the light almost cannot go through the CFRP laminated plates in the wavelength range; the composite transmission rate is 0.

Figure 4b shows that the reflectivity of CFRP laminated plates gradually increases from 8.87% to 9.74%, with the incident wavelength increasing from 1000 nm to 1200 nm. The reflectance of CFRP laminates at 1080 nm is 9.16%.

### 4.2. Ablation Behavior and Ablation Morphologies

Some ablative parameters are defined to describe the ablative behavior of material. Ablative crater depth refers to the maximum ablative depth measured on the surface of CFRP laminates, which can be measured by height scale. The ablation loss mass refers to the mass difference of CFRP laminates before and after laser irradiation, which represents the laser ablation mass loss and can be obtained by high-precision electronic balance. The average power density (laser intensity) is obtained by dividing the laser power by the spot area, which represents the energy of the laser irradiation on the target per unit area in unit time. The linear ablation rate was obtained by dividing the ablation crater depth by the laser irradiation time, which reflected the speed of the ablation process and characterized the ablation boundary regression rate. In order to understand the laser ablation efficiency of samples, the mass ablation rate and thermal ablation rate were calculated. The mass ablation rate was obtained by dividing the ablation loss mass by the irradiation time. The thermal ablation rate was obtained by dividing the ablation loss mass by the irradiation laser energy.

First of all, it is found in the laser irradiation experiment that the burning process gives off a bright light, accompanied by a large amount of black smoke floating in the air. In a long period, after the laser irradiation, CFRP on the surface of the laminated plates before combustion does not stop and will continue to burn, increasing the heat-affected zone (HAZ). CFRP laminated surface laser irradiation of the ablation area in front of the plate above was attached to a large amount of black smoke; the ablation process shows that, in addition to the resin pyrolysis and distillation of carbon fiber, a severe oxidation reaction happened. Ablation morphologies of CFRP laminates (Figure 5) indicates that ablation crater depth increases with incident laser power. The laser ablation damage of carbon fiber on the front surface is mainly sublimation and oxidation reaction.

Ablation morphologies of CFRP laminates under the VHX-100 digital microscope are shown in Figure 6. The CFRP laminate ablation data in the atmospheric environment is shown in Table 1.

As shown in Figure 5, the ablation damage behavior becomes more severe with the high incident laser power. On the one hand, a large amount of heat released by combustion during the ablation process makes the resin outside the laser irradiation area softened and decomposed. On the other hand, carbon fiber’s thermal conductivity is better than resin’s, and the laser irradiation energy is conducted outward along the carbon fiber, expanding the resin pyrolysis area. Compared with 0.88 kW/cm^2^ and 3.54 kW/cm^2^ laser irradiation, obvious ablation pits and loose bulk particles were formed under the high-power density laser irradiation. At the same time, the bulge occurred in the irradiation area under the low power density laser irradiation. The fluffy filaments were exposed, and the filaments were broken. With the power density increase, the ablative behavior shows that the fiber fracture is aggravated, and the carbon fiber’s sublimation reaction and oxidation reaction in the center of the spot are aggravated. The loose carbon block is formed after cooling. At the same time, the temperature at the edge of the spot is low, and the epoxy resin pyrolysis leaves the bare filament. The pyrolysis temperature of epoxy resin is much lower than that of carbon fiber for oxidation and sublimation.

The ablation mass and the mass ablation rate increase with the incident laser power can be seen in Table 2. When the laser power is 250 W, it is found that the ablation pit depth is negative, indicating that the CFRP laminates have a bulging phenomenon. When the laser power reaches 750 W, it is found that the ablation pit depth exceeds the thickness of the CFRP laminate, because with the expansion of the ablation HAZ, the front surface HAZ is higher than the reference plane. The back surface bulge phenomenon occurs, so the ablation crater depth is greater than the thickness of the CFRP laminate. With the increase of laser power density, the ablation rate also increased. However, when the laser power density increases to a specific value, the ablation rate increase is not apparent. It indicates that in a certain amount of irradiation time, the penetration depth of laser ablation is restricted by thermal radiation, heat convection, heat conduction and even the smoke from the burning. Variation of the ablation rate of CFRP laminates with different power densities is shown in Figure 7.

### 4.3. Internal Delamination Damage and Failure Mechanism

To fully understand the effect of laser irradiation, it is necessary to study the internal delamination damage in CFRP laminates. To further elucidate the degree of damage, C-scan was used to obtain delamination. After the test, the specimen was removed from the fixture. The laser irradiation surface was uniformly placed downward when the sample was tested to facilitate the datum measurement. The panels were scanned using an immersion ultrasonic C-scan water tank with 15 MHz transducers in a through-transmission mode, producing a two-dimensional C-scan map of the damaged zone in the laminates. Figure 8 shows the C-Scan images of CFRP laminates irradiated by different laser power.

To comprehensively understand the damage state of CFRP laminates, the spatial geometry of all internal delamination throughout the thickness of the CFRP laminate was accurately identified by the time-of-flight (TOF) technique. The major difference between the C-Scan and the TOF imaging techniques was that the former presented the damage accumulated through the laminate thickness. In contrast, the TOF image delineated the lateral extent and the corresponding depth of delamination on each laminar interface [4].

Figure 9 shows the TOF images of delamination for CFRP laminates irradiated by different laser power. The gray level represents the ultrasonic energy value and the damage characteristics of CFRP laminates with different thicknesses. White represents the largest energy value, indicating the best integrity of CFRP laminates. Black represents the lowest energy value, meaning there is a large amount of delamination damage in this area. Therefore, an ultrasonic C-scan cannot only visualize the expansion of the HAZ under the layer but can also visualize the degree of delamination damage between the layers.

With the increasing incident laser power, the HAZ of CFRP laminates increases obviously, and the heat conduction along the plane leads to different degrees of delamination damage outside the HAZ. The HAZ area is obtained by using image processing software. The HAZ dimensionless area is proposed to further characterize the delamination, as shown in Figure 10.

In Figure 10, A_0_ is the HAZ area on the CFRP laminates laser irradiation surface, and A is the HAZ area on different thicknesses. It is found that the HAZ decreases gradually along the thickness of the laser irradiation surface; the decrease rate of HAZ slows down after 1.2 mm; and the damage degree in the layer reduces significantly. On the one hand, the thermal conductivity along the fiber direction is better than that along the thickness direction. On the other hand, delamination damage prevents heat conduction along the thickness. Therefore, the increase in laser power does not significantly improve the in-plane expansion of the HAZ along the thickness direction. Figure 8 shows the irradiation damage area generated by different laser powers. To evaluate the effect of laser power on ablation damage more directly, the damage area is analyzedd by image processing software. The result is shown in Figure 11. At low laser power density, the HAZ increases obviously with the increase of laser power density. At higher laser power density, there is no significant increase as the laser power density increases, but the HAZ’s damage degree rises.

### 4.4. Temperature Field of CFRP during Laser Irradiation

The center temperature of the CFRP laminated back surface was measured by IR camera, and K type thermocouple is compared and shown in Figure 12. Laser power is 500 W. Irradiation time t is 10 s, so it can be seen that the IR camera is reliable in measuring the CFRP back surface temperature. The temperature curves at different points on the CFRP laminated back surface measured by K type thermocouple are compared and shown in Figure 13. Laser power is 500 W, and irradiation time t is 5 s. The temperature rising rate is largest in the center of the CFRP laminated back surface. The temperature at points 2 and 4 have the same value due to the same location at CFRP laminated back surface. The temperature rise curve of point 2 occurred with slight fluctuations before the temperature reached the highest, which may result in a large amount of gas produced by laser irradiation CFRP laminated. The temperature measuring point not only takes away a lot of heat but also affects the contact of the thermocouple to the back surface of the CFRP laminate. From Figure 13, it can be seen that the time of various points to reach a maximum temperature are larger than the laser irradiation time, because, after the laser irradiation, CFRP laminates will continue to burn, releasing heat in the front surface in the air environment. The CFRP laminated front surface temperature is higher than the back one. After the laser irradiation, the heat is still backward conduction. As a result, the temperature of the rear surface of CFRP laminates continues to rise, and the time to reach the maximum temperature of each point on the rear surface of CFRP laminates is much longer than the laser irradiation time. When the convection dissipation heat loss is enough to offset the heat conduction, the temperature of the rear surface will reach the maximum value. Then the temperature will decrease slowly, making the CFRP laminates’ laser ablation behavior more intense in the air environment.

The images of the temperature field on the CFRP laminated front and back surface irradiated by different power in the air environment are shown respectively with varying times in Figure 14 and Figure 15, with an irradiation time of 20 s. From Figure 14 and Figure 15, it can be seen that the temperature rise rate at the central point of the CFRP laminated front surface can reach 10^6^ °C/s, and the one at the back surface is at the level 100 °C/s. It can be seen that there is a great temperature difference in the center of the front and rear surfaces, indicating that the thermal conductivity of the CFRP laminates is low, and the heat accumulated on the front surface cannot be quickly propagated backwards. The temperature curves of different points at the CFRP laminated rear surface irradiated by different laser power are shown in Figure 16, and the temperature curves of the central point at the CFRP laminated rear surface are shown in Figure 17.

From Figure 16, it can be seen that the temperature of point 2 and point 3 on the back surface does not increase obviously at the beginning, which does not reach the pyrolysis temperature of epoxy resin. Point 2 and point 3 are still in a state of slow heating at the end of the laser irradiation due to the combustion heat release in the laser irradiation area. From Figure 17, the increase of the incident laser power does not significantly increase the temperature rise rate of the center at the rear surface of CFRP laminates. The temperature rise rate is the same below 100 °C, while the temperature rise rate of the center of the rear surface of CFRP laminates shows an obvious difference above 100 °C.

Therefore, the ablation mechanism of CFRP laminates in the air environment under continuous laser irradiation includes the softening and pyrolysis of epoxy resin, the oxidation reaction of carbon fiber and the sublimation reaction of carbon fiber. The combustion heat release of the oxidation reaction is the main reason to aggravate the ablation damage of CFRP laminates. At the same time, it is also the reason that the edge area far away from the center of the rear surface of CFRP laminates keeps slowly heating up for a long time after laser irradiation.

## 5. Conclusions

Laser ablation behaviors of CFRP laminate in the air environment irradiated by different laser power have been investigated experimentally. The main conclusions are as follows:The HAZ of CFRP laminated plates and ablation crater depth increases with incident laser power. The ablation mass and the mass ablation rate increase with the incident laser power. When the laser power is 250 W, it is found that the ablation crater depth is negative, indicating that the CFRP laminates have a bulging phenomenon. When the laser power reaches 750 W, it is found that the ablation crater depth exceeds the thickness of the CFRP laminate.With the increase of the incident laser power, the HAZ area of CFRP laminates increases obviously, and the heat conduction along the plane leads to different degrees of delamination damage outside the HAZ. The dimensionless result of the HAZ area varies along the thickness direction; the HAZ decreases gradually along the thickness of the laser irradiation surface; the decrease rate of HAZ slows down after 1.2 mm; and the damage degree in the layer reduces significantly. At low laser power density, the HAZ increases obviously with increasing the laser power density; at higher laser power density, the HAZ does not significantly increase as the laser power density increases but will increase the damage degree of the HAZ.The IR camera is reliable for measuring the CFRP back surface temperature. The temperature rising rate is largest in the center of the CFRP laminated back surface. The temperature rise rate at the central point of the CFRP laminated front surface can reach 10^6^ °C/s, and the one at the back surface is at the level of 100 °C/s. The thermal conductivity of the CFRP laminates is low.

## Figures and Tables

**Figure 1 polymers-14-05082-f001:**
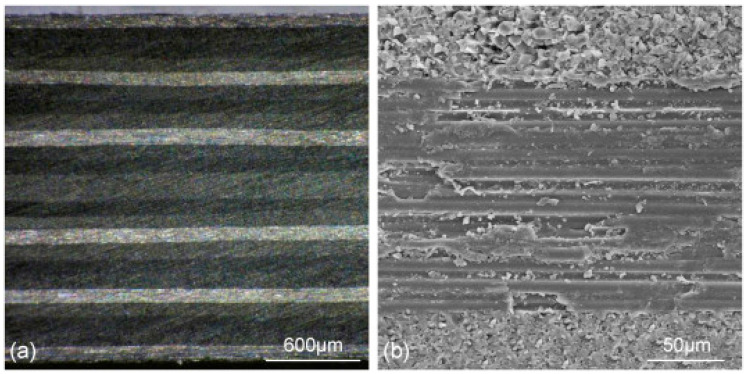
Cross section of CFRP composites at two magnifications (**a**) 20 times; (**b**) 400 times.

**Figure 2 polymers-14-05082-f002:**
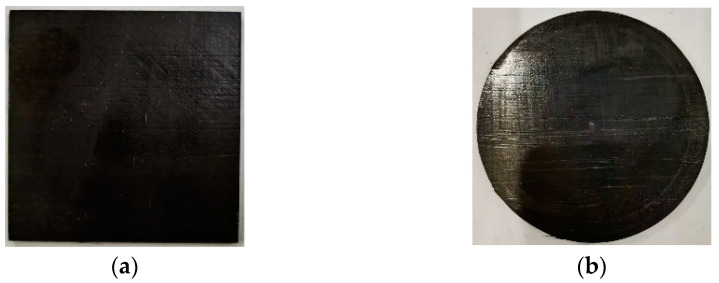
Square and circular CFRP specimens. (**a**) Square CFRP specimens; (**b**) circular CFRP specimens.

**Figure 3 polymers-14-05082-f003:**
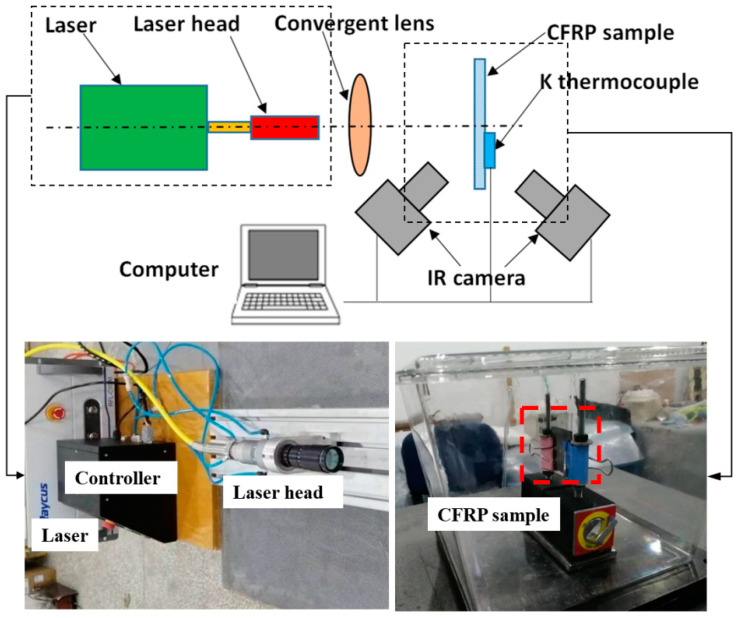
Schematic representation of the experimental setup applied for the investigation of laser irradiation.

**Figure 4 polymers-14-05082-f004:**
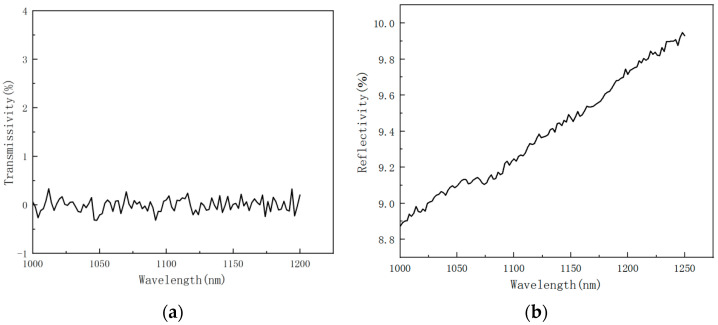
Variation curves of transmittance and reflectance of CFRP laminates as a function of incident wavelength. (**a**) transmission rate; (**b**) reflectance rate.

**Figure 5 polymers-14-05082-f005:**
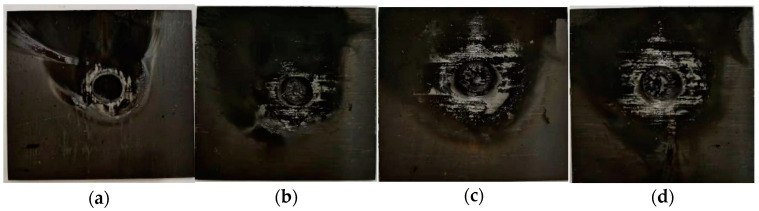
Ablation morphologies of CFRP laminates with different laser power. (**a**) 250 W; (**b**) 500 W; (**c**) 750 W; (**d**) 1000 W.

**Figure 6 polymers-14-05082-f006:**
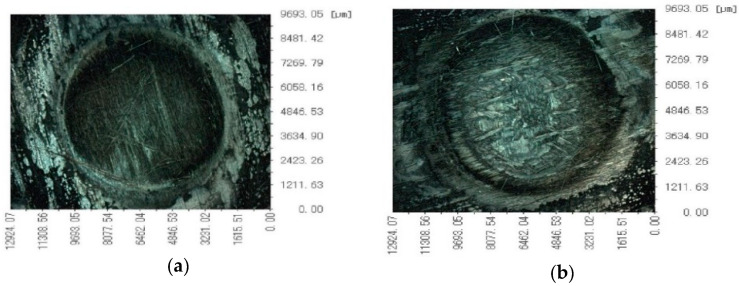
Ablation morphologies of CFRP laminates under VHX-100 digital microscope. (**a**) 250 W; (**b**) 500 W; (**c**) 750 W; (**d**) 1000 W.

**Figure 7 polymers-14-05082-f007:**
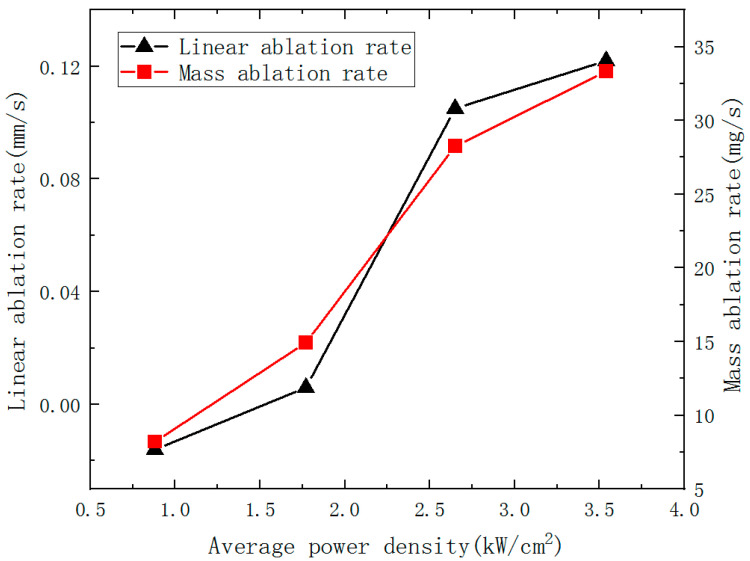
Variation of ablation rate of CFRP laminates with different power density.

**Figure 8 polymers-14-05082-f008:**
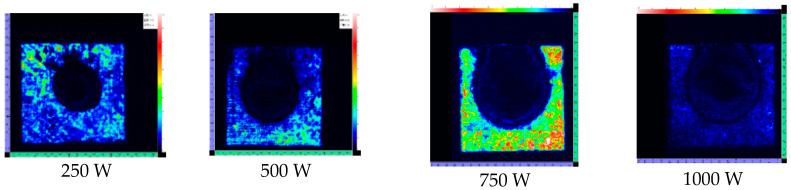
C-Scan images of CFRP laminates at different laser power.

**Figure 9 polymers-14-05082-f009:**
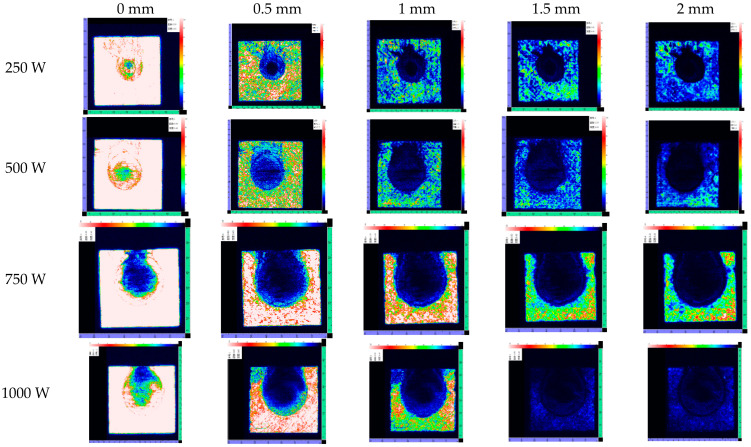
Time-of-flight images showing the distribution of delamination for CFRP laminates irradiated by different laser power.

**Figure 10 polymers-14-05082-f010:**
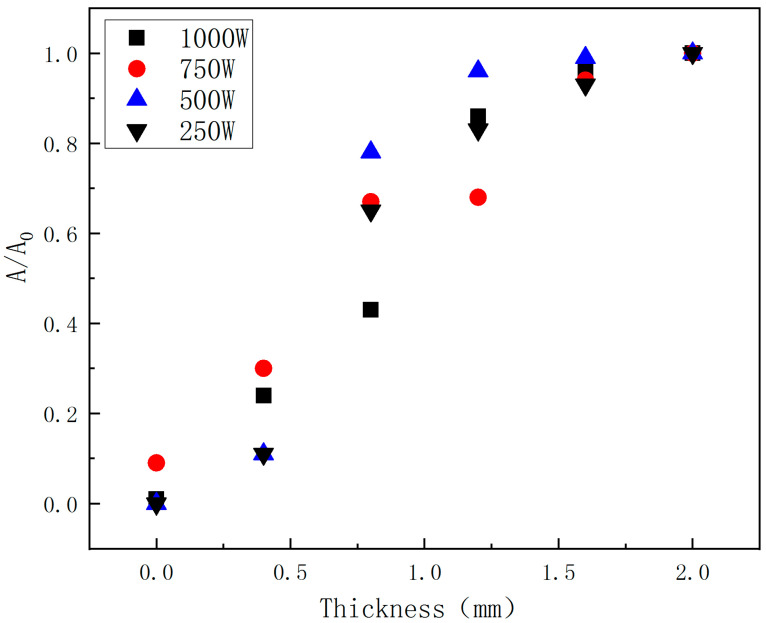
Variation of the HAZ area along the CFRP thickness direction with different laser power.

**Figure 11 polymers-14-05082-f011:**
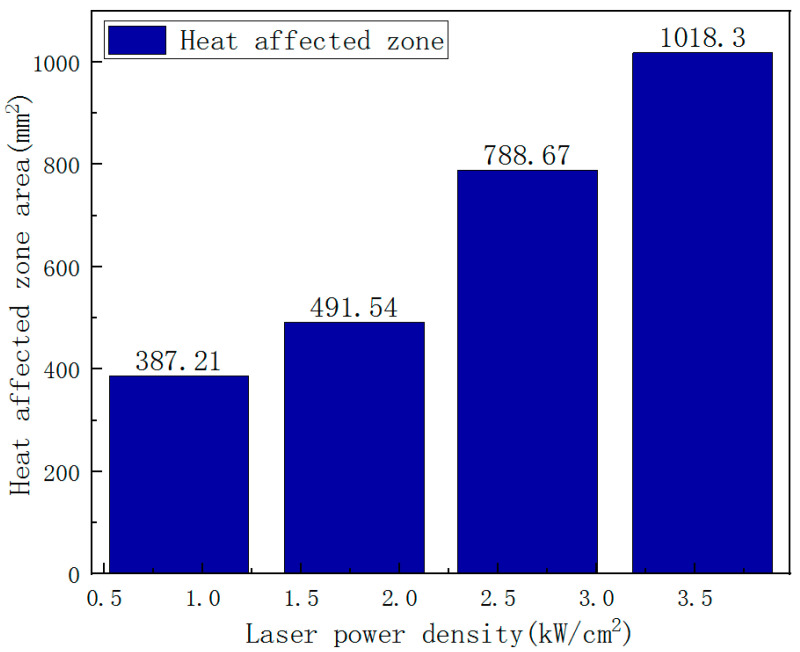
Variation of the HAZ area with different laser power densities.

**Figure 12 polymers-14-05082-f012:**
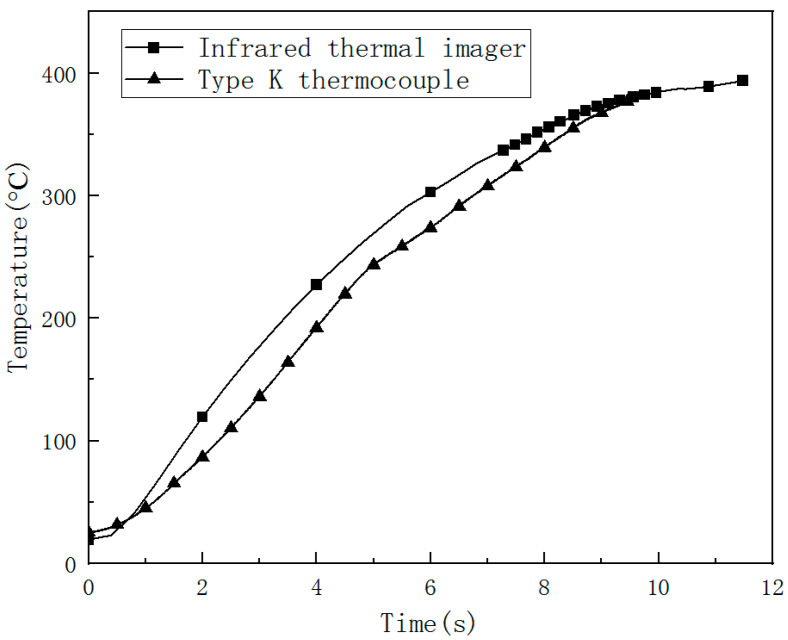
The center temperature of CFRP laminated back surface measured by IR camera and K type thermocouple (500 W, irradiation time 10 s).

**Figure 13 polymers-14-05082-f013:**
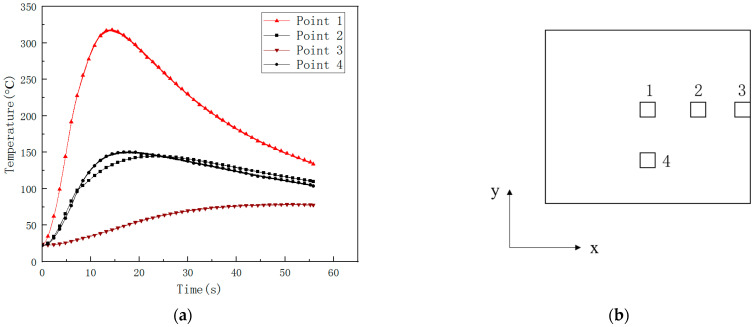
The temperature curves at different points on the CFRP laminated back surface measured by K type thermocouple (500 W, irradiation time 5 s). (**a**) The temperature curves at different points; (**b**) temperature measuring points on the back surface.

**Figure 14 polymers-14-05082-f014:**
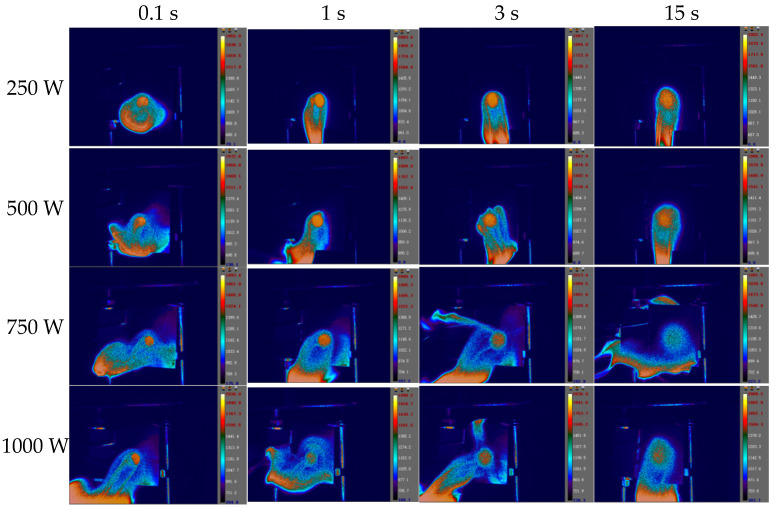
The images of the temperature field on the CFRP laminated front surface irradiated by different laser power in the air environment with different time.

**Figure 15 polymers-14-05082-f015:**
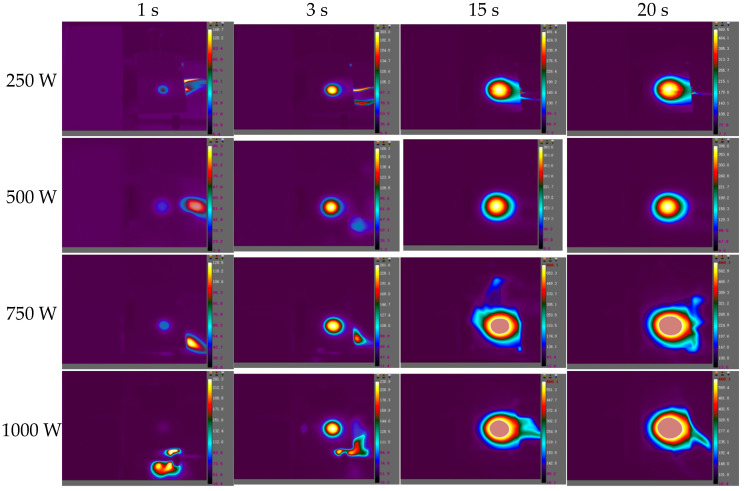
The images of the temperature field on the CFRP laminated back surface irradiated by different laser power in the air environment at different times.

**Figure 16 polymers-14-05082-f016:**
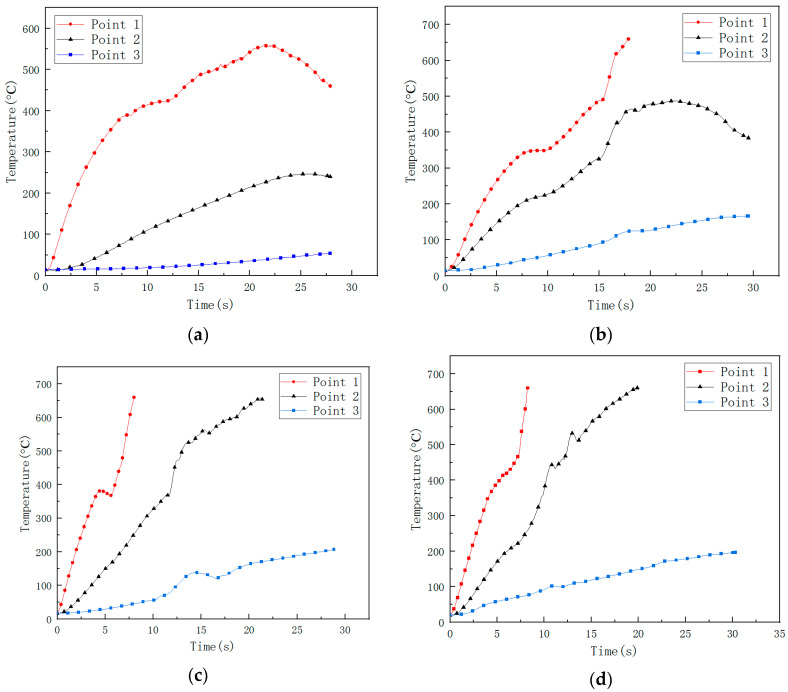
The temperature curve of different points at CFRP laminated rear surface. (**a**) 250 W; (**b**) 500 W; (**c**) 750 W; (**d**) 1000 W.

**Figure 17 polymers-14-05082-f017:**
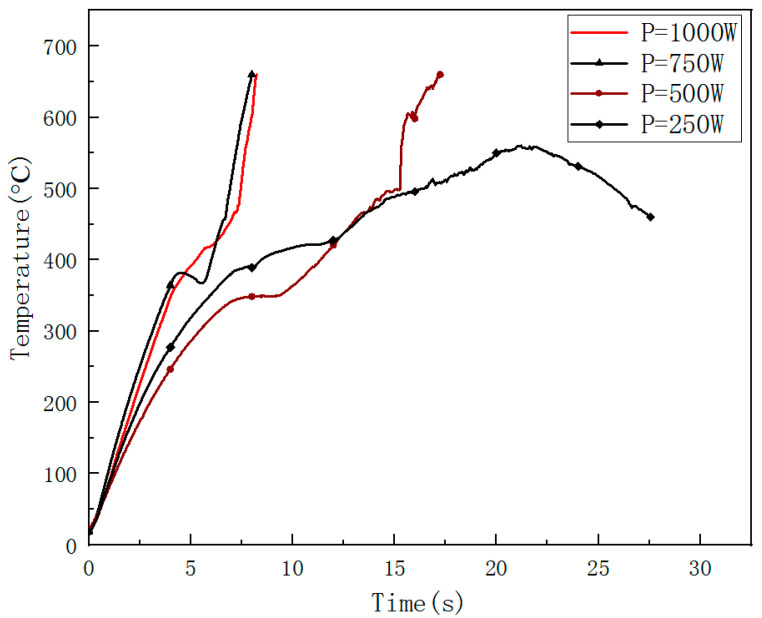
The temperature curve of the central point at CFRP laminated rear surface.

**Table 1 polymers-14-05082-t001:** Material properties of the CFRP [1,2,3,4].

Property	Value
Fiber volume fraction	60%
Longitudinal stiffness, *E*_1_ (GPa)	132
Transverse stiffness, *E*_2_ (GPa)	11
Poisson’s ratio, *v*_12_	0.29
Shear modulus, *G*_12_ (GPa)	5.2
Longitudinal tensile strength, *X*_t_ (MPa)	2178
Longitudinal compressive strength, *X*_c_ (MPa)	1039
Transverse tensile strength, *Y*_t_ (MPa)	24
Transverse compressive strength, *Y*_c_ (MPa)	168
Interlaminar shear strength, *S* (MPa)	81
Density, *ρ* (kg/m^3^)	1600
Longitudinal sound velocity, *c_L_* (mm/μs)	3.212
Shear sound velocity, *c_S_* (mm/μs)	1.472
Bulk sound velocity, *c_B_* (mm/μs)	2.725
Activation energy of carbon fiber E_Af_ (J/kg)	1.55 × 10^5^
Gasification coefficient of carbon fiber Γf	1
Reaction preexponential factor of carbon fiber J_f_ (kg·m^−3^·s^−1^)	4.2 × 10^6^
Activation energy of epoxy matrix E_Ab_ (J/kg)	1.75 × 10^5^
Gasification coefficient of epoxy matrix Γb	1
Reaction preexponential factor of epoxy matrix J_b_ (kg·m^−3^·s^−1^)	6.0 × 10^6^

**Table 2 polymers-14-05082-t002:** The CFRP laminate ablation data in the atmospheric environment.

Power (W)	Power Density (kW/cm^2^)	Crater Depth (mm)	Ablation Mass (g)	Ablation Rate (mm/s)	Mass Ablation Rate (mg/s)	Thermal Ablation Rate (10^−2^ mg/J)
250	0.88	−0.32	0.1642	−0.016	8.21	3.28
500	1.77	0.11	0.2985	0.006	14.93	2.99
750	2.65	2.10	0.5652	0.105	28.26	3.77
1000	3.54	2.44	0.6666	0.122	33.33	3.33
1000	3.54	perforation	—	—	—	—

## Data Availability

Not applicable.

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
