# Peer review of "Experimental Investigation on Ablation Behaviors of CFRP Laminates in an Atmospheric Environment Irradiated by Continuous Wave Laser"

_polymers, 2022, doi:10.3390/polym14235082_

Round 1
Reviewer 1 Report
I found the manuscript difficult to read/understand. Moreover:
- Part 3. Experimental equipment and measuring system is completely missing.
- Table 1 at pag. 7 is actually Table 2 and the text should be modified accordingly.
Other section are not clear as well.
Therefore I cannot recommend the paper for publication in the presen form.
An extensive English language editing is required.
Author Response
Comment 1: I found the manuscript difficult to read/understand. Moreover:
- Part 3. Experimental equipment and measuring system is completely missing.
- Table 1 at page. 7 is actually Table 2 and the text should be modified accordingly.
Response: Thank you for the reviewer’s kind reminder. In the revised manuscript, Experimental equipment and measuring system have been added according to the Reviewer’s suggestion. The text has been modified.
Comment 2: Other section are not clear as well. Therefore I cannot recommend the paper for publication in the present form.
Response: We thank the reviewer’s valuable suggestion. In the revised manuscript, the paper has been improved to show clearly according to the Reviewer’s suggestion.
Comment 3: An extensive English language editing is required.
Response: We thank the reviewer’s valuable suggestion. In the revised manuscript, the English language has been improved according to the Reviewer’s suggestion.
We tried our best to improve the manuscript and made some changes. These changes will not influence the content and framework of the paper. And here, we did not list the changes, but you can see them in Review Mode. We appreciate Editors/Reviewers’ warm work earnestly and hope that the correction will be approved. Once again, thank you very much for your comments and suggestions.

Reviewer 2 Report
Modify the line 20.
Correct the word Unknewn in line 52
Use SI standards for specifying the units in the text line 55, 84, 107, 150, 161, 163, 309 etc
Check the use of Fig. or Figure in the text
Modify the sentences after 89 ( may be a format error title of the figure? Or text?)
Correct the typo and grammatic error in the sentence 102-104
Modify the Figure 12 title ( line 257) : change the word laminated to laminate in the text lines 261, 264, 273, 275, 277, 298 etc
For example, a suggestion is : Figure 14. The images of temperature field on the front surface of the CFRP laminate irradiated by laser at different power and time in ambient condition.
(Change air environment to ambient condition)
Author may specify the use square and circular specimens used in the study.
Based on the results, ablation mechanism of CFRP laminate may be detailed in the text.
Author Response
Comment 1: Modify the line 20; Correct the word Unknewn in line 52; Use SI standards for specifying the units in the text line 55, 84, 107, 150, 161, 163, 309 etc; Check the use of Fig. or Figure in the text; Modify the sentences after 89 ( may be a format error title of the figure? Or text?);Correct the typo and grammatic error in the sentence 102-104; Modify the Figure 12 title ( line 257) : change the word laminated to laminate in the text lines 261, 264, 273, 275, 277, 298 etc; Change air environment to ambient condition
Response: Thank you for the reviewer’s kind reminder. In the revised manuscript, the corresponding changes were made according to the Reviewer’s suggestion.
Comment 2: Author may specify the use square and circular specimens used in the study.
Response: Thank you for the reviewer’s kind reminder. In the revised manuscript, the corresponding changes were made according to the Reviewer’s suggestion.
Comment 3: Based on the results, ablation mechanism of CFRP laminate may be detailed in the text.
Response: Thank you for the reviewer’s kind reminder. In the revised manuscript, the corresponding changes were made according to the Reviewer’s suggestion.
We tried our best to improve the manuscript and made some changes. These changes will not influence the content and framework of the paper. And here, we did not list the changes, but you can see them in Review Mode. We appreciate Editors/Reviewers’ warm work earnestly and hope that the correction will be approved. Once again, thank you very much for your comments and suggestions.

Round 2
Reviewer 1 Report
I thank the authors for taking into account my suggestion. The manuscript is now easier to read although I still believe that English could be improved.
Anyway I wold recommed the paper for publication